# Influence of lower limb isokinetic muscle strength and power on the occurrence of falls in community-dwelling older adults: A longitudinal study

**Cristiane de Almeida Nagata**[1], **Tânia Cristina Dias da Silva Hamu**[2], **Paulo Henrique Silva Pelicioni**[3,4], **João Luiz Quagliotti Durigan**[5]*, **Patrícia Azevedo Garcia**[6]

1 Universidade de Brasília, Programa de Pós-Graduação em Educação Física, Brasília, DF, Brazil,
2 Laboratório de Pesquisa em Musculoesquelética, Universidade Estadual de Goiás, Goiânia, GO, Brazil,
3 School of Health Sciences, University of New South Wales, Randwick, NSW, Australia, 4 Neuroscience Research Australia, University of New South Wales, Randwick, NSW, Australia, 5 Universidade de Brasília, Laboratory of Muscle and Tendon Plasticity, Programa de Pós-Graduação em Educação Física, Brasília, DF, Brazil, 6 Universidade de Brasília, Programa de Pós-Graduação em Ciências da Reabilitação, Brasília, DF, Brazil

☯ These authors contributed equally to this work.
* joaodurigan@gmail.com

**Data Availability Statement:** All relevant data are within the manuscript and its Supporting Information files.

## Abstract

### Introduction

Previous studies have highlighted the association between lower limb muscle strength and falls in older adults. However, a comprehensive understanding of the specific influence of each lower limb muscle group on fall occurrences remains lacking.

### Objective

This study aimed to investigate the impact of knee, ankle, and hip muscle strength and power on falls in older adults, with the goal of identifying which muscle groups are more predictive of fall risk in this population.

### Methods

This longitudinal observational study enrolled 94 community-dwelling older adults. Muscle strength and power of the ankle's plantiflexors and dorsiflexors, knee flexors and extensors, and hip flexors, extensors, adductors, and abductors were assessed using a Biodex System 4 Pro® isokinetic dynamometer. Fall occurrences were monitored through monthly telephone contact over a year.

### Results

Participants, with a median age of 69 years (range 64–74), included 67% women, and 63.8% reported a sedentary lifestyle. Among them, 45,7% of older adults were classified as fallers. Comparative analyses revealed that non-fallers displayed significantly superior

**Funding:** This study was partially funded by the Postgraduate Department from the University of Brasília, https://dpg.unb.br/ (SEI N˚ 23106.102043/2017-01 author CAN), the Deanery of research from the State University of Goiás, https://ueg.br/prp/ (CCB 01/2018 author TCDSH), "Fundação de Apoio a Pesquisa do Distrito Federal" (FAPDF), https://www.fap.df.gov.br/ (grant number 00193-00000866/2023-6 author PAG, 00193.00000773/2021-72 author JLQD, 00193.00000859/2021-3 author JLQD; 00193.00001222/2021-26 author JLQD, 00193-00001261/2021-23 author JLQD), and the National Council for Scientific and Technological Development (CNPq), https://www.gov.br/cnpq/pt-br (process numbers 309435/2020-0 and 310269/2021 author JLQD). The funders had no role in the study design, data collection and analysis, decision to publish, or preparation of the manuscript. There was no additional external funding received for this study.

**Competing interests:** None.

isokinetic muscle strength in the hip abductors and adductors, along with higher muscle power in the hip abductors, hip flexors, and knee flexors compared to fallers. Multivariate logistic regression analysis indicated that a 1 Nm/Kg increase in hip abductor strength reduced the chance of a fall by 86.3%, and a 1 Watt increase in hip flexor power reduced the chance of a fall by 3.6%.

## Conclusion

The findings indicate that hip abductor strength and hip flexor power can be considered protective factors against falls in independent older adults in the community. These findings may contribute to developing effective fall-prevention strategies for this population.

## Introduction

The occurrence of falls in older adults is a global health problem [1]. Investigations into the causes of and risk factors for falls in older adults are critical because these findings could help in the creation of rehabilitation and fall prevention programs [1–3]. Muscle function is one of the essential factors for balance recovery and fall avoidance [2, 3]. Muscle function comprises the ability to generate and control movement, which is important to maintain postural control, an essential factor for an individual to minimize eventual balance disturbances and to avoid falls [2, 3]. Muscle function can be quantified by analyzing parameters related to muscle contraction, such as peak torque, which indicate the maximal muscle strength and power, representing the ability of the muscle to exert a large amount of force at high speed [2].

With advancing age, there is a decline in muscle strength and power [4–9]. Muscle strength decreases by 1.8 to 2.0% per year and power by 3.2 to 3.7% per year in older adults (age range 65–89 years) [4]. According to Frontera et al. [10], in sedentary men with initial mean age 65.4 years-old (age range 60–72 years), there was a 20 to 30% reduction in the isokinetic strength of knee flexors and extensors at low and high speeds in twelve years. Lanza et al. [11] demonstrated that older adults produced 26% less concentric torque and power in the dorsiflexors and 32% less in the knee extensors than young adults.

Changes in muscle strength and muscle power have been related to a higher occurrence of falls in older adults [12–14]. Clynes et al. [12] demonstrated that individuals with sarcopenia reported a greater number of falls in the previous year. Scott et al. [13], in a prospective study, showed that older adults with sarcopenia were more likely to report falls in a one-year follow-up; and Moreland et al. [14], in a meta-analysis, indicated that lower limb weakness is a clinically important risk factor for falls.

While the association between muscle strength, muscle power, and falls has been demonstrated in previous studies [2, 3, 12–24], the specific influence of each lower limb muscle group's strength and power on the occurrence of falls in older individuals remains unclear. Some studies suggest that ankle muscle strength and power are more closely associated with falls in older adults [2, 15–20, 24]. However, other research indicates that knee flexors [21, 22], knee extensors [21–23], hip flexors [3], hip extensors [3, 22, 23], hip abductors [3, 21], and hip adductors [3] also play crucial roles in this outcome.

Notably, except for Daubney et al. [19], none of the cited studies have comprehensively examined the influence of all eight muscle groups in the lower limbs on falls in older adults. Additionally, most of the available studies did not evaluate muscle strength and power using the isokinetic dynamometer [16, 17, 19, 21] (considered the gold standard for assessing muscle

strength [25]), and they often focused on isolated muscle groups or a subset of the lower limb muscles assessment [3, 15–18, 20, 22], which may not provide a complete understanding of overall muscle function.

Therefore, understanding the simultaneous association between all eight muscle groups of the lower limbs and falls in older adults is crucial for developing an effective fall prevention program. Our study addresses this gap by investigating by isokinetic dynamometer the influence of ankle plantarflexors, ankle dorsiflexors, knee flexors, knee extensors, hip flexors, hip extensors, hip abductors and hip adductors muscle strength and muscle power on falls in older adults, aiming to determine which muscle groups are more predictive of fall risk in this population. This comprehensive approach may offer insights for developing targeted and effective strategies in fall prevention and rehabilitation. Based on the existing literature[2, 3, 12–24], our hypothesis is that our findings will align with previous findings[2, 15–20, 24], indicating that ankle muscles will be more predictive of the occurrence of falls in older adults.

## Materials and methods

### Study design

A longitudinal observational study was approved by the Research Ethics Committee of the Faculty of Ceilândia at the University of Brasília (CAAE 70241817.7.0000.8093). All participants signed the written informed consent form. Participants were recruited through printed flyers distributed in the community. The data were collected between October 2017 and February 2021. Inclusion criteria were older adults ($\geq$60 years old), showing an independent gait, and no marked cognitive impairment examined by the Mini-Mental State Examination (MMSE) [26] (cut-off point of 20 points for illiterate individuals; 25 points for people with 1 to 4 years of schooling; 26.5 for 5 to 8 years of schooling; 28 for those with 9 to 11 years of schooling; and 29 for more than 11 years of schooling). Exclusion criteria were sequelae of severe cardiorespiratory or neurological diseases, a history of fractures, or recent surgeries ($<$6 months) in the lower limbs. In addition, participants who failed to perform the tests and missed the phone calls for prospective analysis of falls for more than two months ($<$ 20% of loss) were excluded.

The sample size was calculated using GPower 3.1.5 software. Based on our preliminary findings, which identified an odds ratio (OR) of 0.438 for the association between muscle strength and the occurrence of falls and considering a desired power of 80% and an alpha error of 0.05, the estimated sample size was determined to be 83 older adults. We planned to include an additional 10% of participants, bringing the total number of participants to 94 (Fig 1). This adjustment was intended to account for any potential dropouts or withdrawals during the research.

For demographic and physical-functional data, information regarding the age, sex, nutritional status, five time sit-to-stand test, usual walking speed, and physical activity levels of participants was collected. Nutritional status was determined by the Body Mass Index and classified according to Lipschitz et al. [27] To determine physical activity levels, participants were asked how many minutes per week they practiced moderate to vigorous exercise and were classified as active ($\geq$150 minutes of activity per week) or sedentary [28]. To assess walking speed, a 4-meter distance was marked, and participants were instructed to walk at their usual pace. The time to complete the test was recorded in two attempts, with the shorter time to complete the walking assessment considered for scoring. The usual walking speed (m/s) was then calculated by dividing the distance covered in the test (4 meters) by the shortest time recorded for that course in seconds [29–31]. In the five-time sit-to-stand test, participants were asked to rise from a chair, starting from a seated position, with arms crossed over the

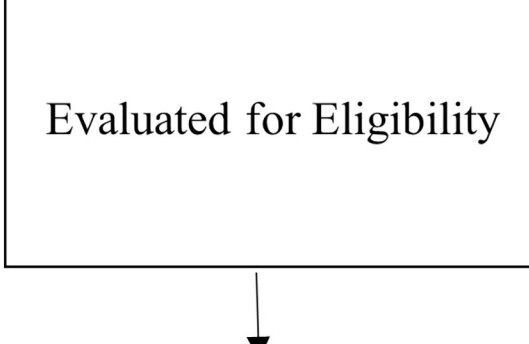

**Fig 1. Flow diagram of participants.** Notes: MMSE = Mini-Mental State Examination.

chest, repeating the movement five times as quickly as possible. The elapsed time in seconds was recorded as a performance measure [29–31]. These parameters were considered potential confounders in the analyses.

**Occurrence of falls.** The dependent variable of this study was the assessment of prospective falls, collected during one year. The occurrence of falls was collected through telephone calls once a month for a year, when participants were asked if "In the previous month, did you have any falls? If yes, how many?". Those who responded that they had experienced a fall were

also asked to report the reason, the location where the fall occurred, whether it caused any injuries, and whether they needed help to get up. Participants who did not fall once during the 12 months of follow-up were classified as "non-fallers". Participants who fell once or more were classified as "fallers". Falls were defined as unexpected and unintentional events that lead the individual to rest on the floor or at a lower level than they were [32].

**Lower limb muscle strength and muscle power.** Lower limb muscle strength and muscle power were assessed as independent variables for the following muscle groups: knee extensors, knee flexors, plantiflexors, dorsiflexors, hip flexors, hip extensors, hip adductors, and hip abductors. The strength index used in the analysis was the peak torque per body weight (Nm/Kg), and the power index was the average power in Watts. Peak torque is the maximum force produced during a muscle contraction and muscle power is the ability of the muscle to exert a large amount of force at high speed [2]. We used the Biodex System 4 Pro® isokinetic dynamometer (Biodex Medical Systems Inc.), a reliable mode of muscle strength and power assessment (ICC = 0.99 to 1.0) [33]. For this assessment, participants were instructed not to practice physical exercise and not to drink energetic or alcoholic beverages within 24 hours before the laboratory visit. The equipment was calibrated before the start of each testing session according to the manufacturer's instructions.

Before the assessment, a warm-up was performed using a cycle ergometer for 5 minutes. To familiarize participants with the evaluation procedures, attempts were performed with three submaximal repetitions at the same test speeds [25, 34, 35]. The order of evaluation was randomized by drawing opaque envelopes containing the names of the joints. Measurements were only collected for the dominant limb (determined by the Waterloo Questionnaire [36]), using concentric contractions, constant angular velocities, and careful positioning. Participants were instructed to keep their knee extended during the hip flexion and hip extension tests. They were also instructed to keep their toes forward and not to flex the knee [2] during the hip abduction and hip adduction tests. Muscle strength was evaluated at 60°/s with 5 repetitions. Muscle power was evaluated at 120°/s with 6 repetitions for ankle plantiflexors and dorsiflexors, 120°/s with 15 repetitions for hip flexors, hip extensors, hip adductors, and hip abductors, and 180°/s with 15 repetitions for knee extensors and knee flexors (Fig 2). During the tests, participants were verbally encouraged to produce their maximum torque and a 2-minute rest period was given between sets.

**Statistical analysis.** The continuous data were analyzed descriptively using central tendency (mean) and variability (standard deviation) measures. The categorical data are presented in frequency and percentage. The normal distribution of the data was identified using the Kolmogorov-Smirnov test. To compare groups, student-t or U Mann-Whitney tests for independent samples were used for parametric and non-parametric data, respectively.

Multivariate logistic regression (backward LR method) analysis was used to determine the association between independent and dependent variables and, in this way, to verify the parameters of isokinetic muscle strength and power that contributed to falls. To obtain more reliable results, and truly consider the analysis of all muscle groups, we opted for the method of listwise deletion. The identification of "fallers" was categorized as "1" in the logistic regression analyses. Odds ratios (OR) were calculated for each explanatory variable with 95% confidence intervals. The models respected the postulates of a multivariate logistic regression: homogeneity, homoscedasticity, absence of collinearity, and normality of residues. Multicollinearity was considered with a tolerance of <0.1 and variance inflation factor (VIF) > 10. A level of significance of 5% was considered. Statistical analyses were processed using the Statistical Package for Social Sciences (SPSS), version 22.0.

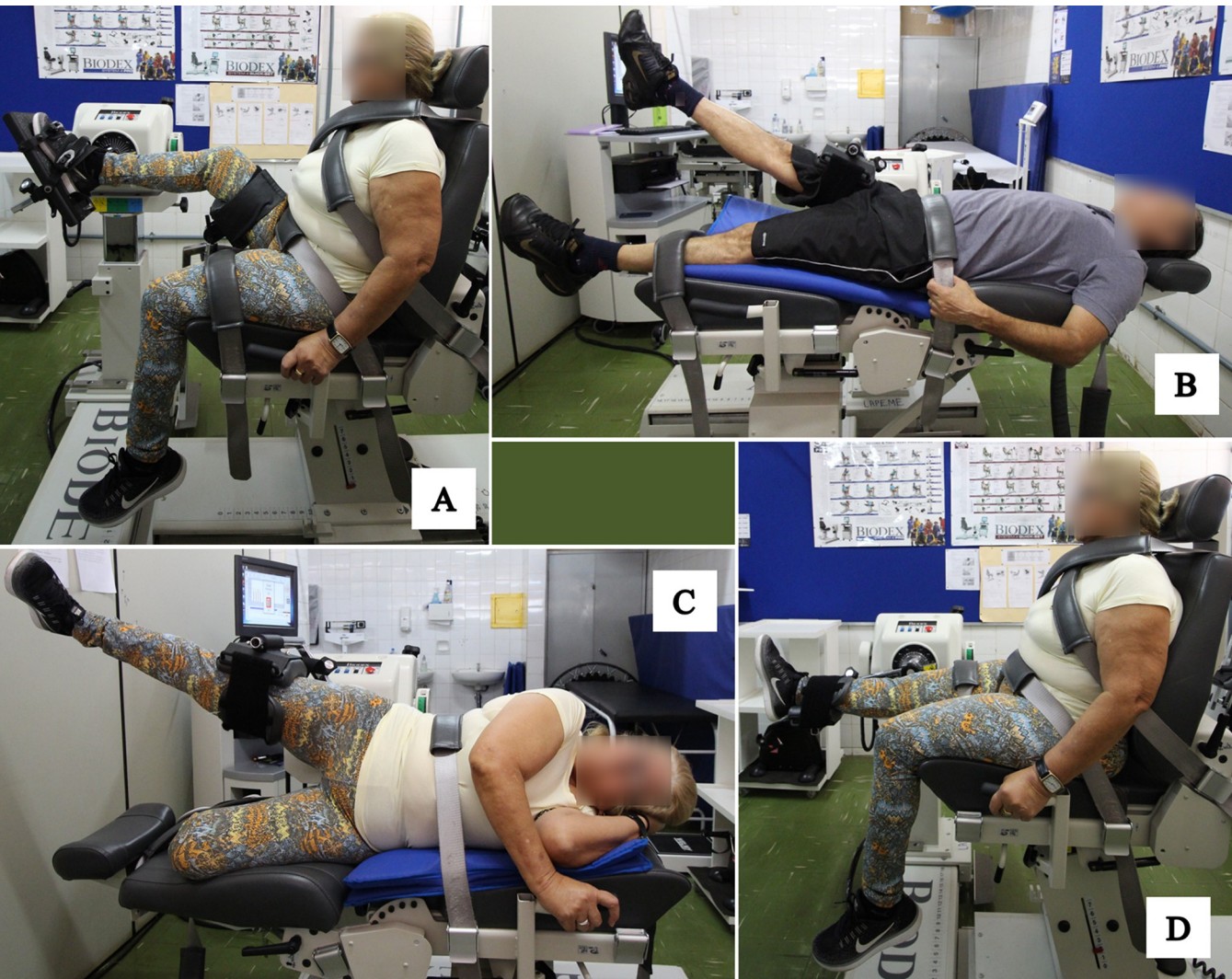

**Fig 2. Lower limb muscle strength and muscle power assessment.** (A) Ankle plantiflexion and dorsiflexion. (B) Hip flexion and extension. (C) Hip abduction and adduction; (D) Knee flexion and extension.

## Results

Ninety-four community-dwelling older adults participated in all phases of the evaluation (Fig 1). Forty-three older adults (45.7%) were classified as fallers. The characteristics of the participants are presented in Table 1.

There were no statistical differences in age, sex, physical activity level, and nutritional status between fallers and non-fallers (Table 1). However, the fallers showed worse functional performance than non-fallers in the sit-to-stand (Cohen D effect size of 0.40) and walking speed tests (Cohen D effect size of 0.53) (Table 1).

The data showed that non-fallers presented significantly better hip abductor and hip adductor isokinetic muscle strength, and hip abductor, hip flexor, and knee flexor muscle power than fallers (Table 2).

The multivariate logistic regression analysis showed that an increase of 1 Nm/Kg in hip abductor strength reduces the chance of a fall in older adults by 86.3% (Table 3). Moreover, the model correctly classified 80.4% of non-fallers and 46.5% of fallers. An increase of 1 Watt

**Table 1. Characteristics of the study sample.**

| Variables | Total Sample (n = 94) | Fallers (n = 43) | Non-fallers (n = 51) | p-value |
|---|---|---|---|---|
| Age (years) [a] | 69 (64–74) | 69 (64–75) | 68 (65–47) | 0.954 |
| Sex[b] | | | | |
| Female | 63 (67) | 33 (52.4) | 30 (47.6) | 0.066 |
| Male | 31 (33) | 10 (32.3) | 21 (67.7) | |
| Nutritional status[b] | | | | |
| Underweight | 8 (8.5) | 4 (50) | 4 (50) | 0.515 |
| Eutrophic | 41 (43.6) | 16 (39) | 25 (61) | |
| Overweight | 45 (47.9) | 23 (51.1) | 22 (48.9) | |
| Physical Activity[b] | | | | |
| Active | 34 (36.2) | 19 (55.9) | 15 (44.1) | 0.138 |
| Sedentary | 60 (63.8) | 24 (40) | 36 (60) | |
| **Functional Performance** | | | | |
| Sit-to-Stand (s) [a] | 10.06 (8.0–11.43) | 10.37 (8.96–12.78) | 9.37 (7.83–10.81) | 0.009 |
| Walking Speed (m/s) [c] | 0.98 (0.19) | 0.92 (0.23) | 1.00 (0.16) | 0.035 |

Notes

[a]Median (interquartile range 25% - 75%)

[b]Frequency (percentage)

[c]Mean (standard deviation)

**Table 2. Isokinetic muscle strength and muscle power data from fallers and non-fallers.**

| Variables | Total Sample (n = 94) | Fallers (n = 43) | Non-fallers (n = 51) | Mean Difference (95% CI) | p-value |
|---|---|---|---|---|---|
| **Muscle Strength (Nm/Kg)** | | | | | |
| Ankle Plantiflexors [a] | 0.44 (0.18) | 0.40 (0.20) | 0.46 (0.18) | 0.06 (-0.02; 0.14) | 0.126 |
| Ankle Dorsiflexors [b] | 0.24 (0.20–0.26) | 0.23 (0.19–0.25) | 0.24 (0.20–0.28) | - | 0.130 |
| Knee Flexors[a] | 0.63 (0.21) | 0.59 (0.25) | 0.65 (0.19) | 0.06 (-0.03; 0.15) | 0.223 |
| Knee Extensors[a] | 1.35 (0.42) | 1.25 (0.51) | 1.37 (0.37) | 0.12 (-0.06; 0.30) | 0.183 |
| Hip Flexors[a] | 0.73 (0.27) | 0.67 (0.34) | 0.76 (0.23) | 0.09 (-0.03; 0.21) | 0.139 |
| Hip Extensors[b] | 0.89 (0.62–1.13) | 0.81 (0.46–1.07) | 0.94 (0.69–1.29) | - | 0.130 |
| Hip Abductors[a] | 0.86 (0.25) | 0.78 (0.27) | 0.90 (0.24) | 0.13 (0.02; 0.23) | 0.018* |
| Hip Adductors[a] | 0.74 (0.28) | 0.66 (0.29) | 0.79 (0.28) | 0.13 (0.16; 0.25) | 0.027* |
| **Muscle Power (Watts)** | | | | | |
| Ankle Plantiflexors [b] | 15.80 (9.62–23.47) | 14.10 (8.15–21.30) | 16.40 (10.0–25.90) | - | 0.141 |
| Ankle Dorsiflexors [b] | 11.90 (8.97–13.92) | 11.80 (8.35–14.12) | 11.90 (9.30–13.70) | - | 0.735 |
| Knee Flexors[a] | 37.57 (19.23) | 32.77 (20.51) | 40.81 (17.64) | 8.04 (0.23; 15.86) | 0.044* |
| Knee Extensors[a] | 80.41 (31.01) | 73.28 (35.00) | 85.22 (26.67) | 11.94 (-0.70; 24.59) | 0.064 |
| Hip Flexors[a] | 30.88 (15.42) | 26.13 (15.64) | 34.25 (14.38) | 8.12 (1.97; 14.28) | 0.010* |
| Hip Extensors[b] | 32.75 (14.75–52.87) | 28.70 (11.40–45.70) | 34.50 (19.40–53.80) | - | 0.420 |
| Hip Abductors[b] | 38.70 (29.17–54.50) | 34.20 (23.60–44.70) | 42.10 (34.45–56.77) | - | 0.021* |
| Hip Adductors[b] | 17.75 (7.10–31.15) | 11.80 (2.70–25.50) | 20.15 (9.15–34.35) | - | 0.059 |

[a]Mean (Standard deviation). student-t test for independent samples.

[b]Median (25th - 75th). U Mann-Whitney test for independent samples.

[c]Frequency (Percentage) Chi-Square

*p<0.05.

Note: 95% CI was not presented for variables with non-normal distribution.

**Table 3. Multivariate logistic regression (backward LR method).**

| | | | Standardized Coefficients | Individual significance |
|---|---|---|---|---|
| **Muscle Strength (Nm/Kg)** | | | Exp (B) (95% IC) | p-value |
| **Falls Occurrence** | Regression Step 1 | Ankle Plantiflexors | 0.461 (0.012; 18.252) | 0.680 |
| | | Ankle Dorsiflexors | 1.633 (0.002; 1214.8) | 0.884 |
| | | Knee Flexors | 2.511 (0.045;140.85) | 0.654 |
| | | Knee Extensors | 2.428 (0.241; 24.412) | 0.451 |
| | | Hip Flexors | 1.326 (0.053; 33.048) | 0.864 |
| | | Hip Extensors | 1.300 (0.269; 6.280) | 0.744 |
| | | Hip Abductors | 0.069 (0.003; 1.635) | 0.098 |
| | | Hip Adductors | 0.159 (0.012; 2.149) | 0.166 |
| | **Regression Step 8** | **Hip Abductors** | 0.137 (0.025; 0.745) | 0.021* |
| **Muscle Power (Watts)** | | | Exp (B) (95% IC) | p-value |
| **Falls Occurrence** | Regression Step 1 | Ankle Plantiflexors | 0.974 (0.900; 1.053) | 0.508 |
| | | Ankle Dorsiflexors | 1.062 (0.913; 1.236) | 0.434 |
| | | Knee Flexors | 0.989 (0.946; 1.034) | 0.617 |
| | | Knee Extensors | 1.011 (0.982; 1.040) | 0.461 |
| | | Hip Flexors | 0.966 (0.909; 1.026) | 0.256 |
| | | Hip Extensors | 1.023 (0.988; 1.060) | 0.197 |
| | | Hip Abductors | 0.977 (0.922; 1.034) | 0.418 |
| | | Hip Adductors | 0.984 (0.927; 1.044) | 0.592 |
| | **Regression Step 8** | **Hip Flexors** | 0.964 (0.936; 0.992) | 0.013* |

*$p < 0.05$.

in hip flexor power reduces the chance of a fall in older adults by 3.6% (Table 3). The model correctly classified 64% of non-fallers and 50% of fallers.

## Discussion

The present study analyzed the association between lower limb muscles and the occurrence of falls in community-dwelling older adults. The findings indicated that an increase of 1 Nm/Kg in hip abductor strength reduces the chance of a fall in older adults by 86.3%, and an increase of 1 Watt in hip flexor power reduces the chance of a fall in older adults by 3.6%. The results point to a preliminary influence of the strength of these key lower limb muscle groups as a protective factor for the occurrence of falls in community-dwelling older adults.

Our findings indicated that fallers exhibited worse functional performance than non-fallers in the sit-to-stand and walking speed tests. These findings align with various prior studies [37–42] that emphasized the significant association between slower walking speed and occurrences of falls in older adults. These results can be explained by the impact of aging on the gait of older individuals [43], characterized by changes in three-dimensional kinetics and kinematics, including reduced movement of the pelvis in the frontal and transverse planes, prolonged hip adduction, and increased knee extension peak during the support phase (hyperextension). These alterations, coupled with reduced ankle plantar flexion during the terminal support phase and decreased knee range of motion compared to younger individuals, contribute to the decrease in walking speed [43]. Additionally, previous studies have also highlighted the association between poorer performance in the sit-to-stand test and a higher risk of falls in older adults [44–46]. This relationship can be attributed to the influence of aging on this function [4, 6], as evidenced by the greater difficulty that older individuals face when rising from a chair compared to younger counterparts [6, 8].

Comparisons between older adults who experienced a fall and those who did not revealed that non-fallers had significantly better isokinetic muscle strength in the hip abductors and adductors, as well as higher muscle power in hip abductors, hip flexors, and knee flexors compared to fallers. This finding is somewhat surprising, considering that several previous studies suggested that ankle muscle strength and power are more closely associated with falls in older adults [2, 15–20, 24]. On the other hand, Morcelli et al. [3] had already demonstrated that older fallers had significantly lower strength in hip extension, abduction, and adduction, as well as lower power in hip flexion, extension, and abduction compared to non-fallers, which is consistent with our findings.

The association between hip abductor strength and the occurrence of falls has already been demonstrated in previous studies [3, 21, 47], showing that older adults who experienced previous falls also presented weak hip abduction [3, 47]. Other studies also showed that hip abductors are involved in controlling postural sway in older adults [48], contributing to the maintenance of static and dynamic balance [49] and being critical for balance and mobility function [50]. In addition, the hip abductor muscles are particularly important in the stance phase of walking, when these muscles are required to provide lateral support at the hip joint and to generate frontal plane stability of the hip, avoiding contralateral pelvis drop, and thereby maintaining the proper alignment between the trunk and lower extremities [21, 51–53]. Accordingly, the decline in hip strength abduction may impact mediolateral stability, which is vital in maintaining the postural control of the trunk [3], in order to develop rapid and appropriate compensatory postural adjustments to prevent falls and fall-related injuries.

We also observed that an increase in hip flexor power reduces the chance of the occurrence of falls in older adults. This finding is in agreement with the results found by Morcelli et al. [3], who indicated that older fallers presented lower power during hip flexion movements. Orr et al. [54] found that there was insufficient evidence of the contribution of muscle weakness to postural instability in healthy older adults, arguing that muscle power could be more predictive of falls than strength alone, since during a disturbance in postural control (e.g., reactive balance), an individual needs to develop strength quickly to regain balance. Therefore, the power and speed of hip flexor muscle contraction could greatly influence the occurrence of falls since, during balance perturbations, older adults utilize the step strategy as their last resort from a mechanistic point of view, and if the perturbation is forward this involves rapid contraction of the hip flexors. Furthermore, hip flexors are involved in raising the lower limb during the swing phase of gait, thus allowing sufficient toe clearance, which is important for avoiding falls [51, 55].

Contrasting with findings from prior studies [2, 15, 24, 16–23], our observations highlight a notable absence of significant associations between the strength and power of muscle groups in the knee and ankle joints and occurrences of falls in older adults. This discrepancy can be attributed to the fact that previous research did not include a comprehensive analysis of all eight muscle groups in the lower limbs and did not utilize the isokinetic dynamometer, acknowledged as the gold standard for assessing muscle strength [25]. These methodological limitations likely played a role in the disparities among study outcomes, posing a challenge in establishing coherent relationships between distinct muscle groups and the incidence of falls. Additionally, most research has focused exclusively on assessing the musculature of the ankle and/or knee, disregarding the analysis of hip muscles. This may explain why the importance of these muscles in functional performance and fall prevention in older adults has not been fully elucidated.

To our knowledge, this is the first study that analyzed the association between isokinetic strength of the lower limb muscle groups and prospective falls in community-dwelling older adults. However, the present study exhibits some limitations that could affect the

interpretation of the results. Firstly, the relationship detected between muscle strength and the occurrence of falls did not reflect a causal relationship. However, it could explain the performance of older adults and indicate targets for potential interventions and topics for future clinical trials. Another possible limitation of our study is that our sample consisted of a large majority of robust and healthy older adults. Despite this, almost half (45.7%) of the older adults fell at least once in one year. This fact demonstrates the importance of implementing programs to prevent falls for older adults who do not yet present significant functional limitations, and to increase their functional independence.

Despite the limitations mentioned above, we demonstrated that hip abductor strength and hip flexor power could be the key muscle groups involved in the occurrence of falls in community-dwelling older adults. Thus, the results of our study may assist in designing time-efficient prevention and intervention strategies for preventing falls in older people. The adherence of healthy older adults to exercise programs can be challenging, with prescribed exercise duration being the strongest determinant of adherence in this population [56]. Therefore, reducing the training time needed by focusing on the core muscles for these functional tasks could be a valid strategy. Further studies are needed to assess whether intervention programs that include strength training for hip abductors and power training for hip flexor muscles lead to clinically significant decreases in falls in community-dwelling older adults.

## Conclusion

Hip abductor muscle strength and hip flexor muscle power measured through isokinetic assessment are positively associated with prospective falls in community-dwelling older adults. Thus, maintaining these levels of muscle strength and power might be a strategy to reduce future falls, alleviating public health concerns and saving billions in public health costs per year.

## Supporting information

**S1 Dataset.**
(SAV)

## Author Contributions

**Conceptualization:** Cristiane de Almeida Nagata, Tânia Cristina Dias da Silva Hamu, João Luiz Quagliotti Durigan, Patrícia Azevedo Garcia.

**Data curation:** Cristiane de Almeida Nagata, Tânia Cristina Dias da Silva Hamu, João Luiz Quagliotti Durigan, Patrícia Azevedo Garcia.

**Formal analysis:** Cristiane de Almeida Nagata, Tânia Cristina Dias da Silva Hamu, João Luiz Quagliotti Durigan, Patrícia Azevedo Garcia.

**Funding acquisition:** Cristiane de Almeida Nagata, Tânia Cristina Dias da Silva Hamu, João Luiz Quagliotti Durigan, Patrícia Azevedo Garcia.

**Investigation:** Cristiane de Almeida Nagata, Tânia Cristina Dias da Silva Hamu, João Luiz Quagliotti Durigan, Patrícia Azevedo Garcia.

**Methodology:** Cristiane de Almeida Nagata, Tânia Cristina Dias da Silva Hamu, João Luiz Quagliotti Durigan, Patrícia Azevedo Garcia.

**Project administration:** Cristiane de Almeida Nagata, Tânia Cristina Dias da Silva Hamu, João Luiz Quagliotti Durigan, Patrícia Azevedo Garcia.

**Resources:** Cristiane de Almeida Nagata, Tânia Cristina Dias da Silva Hamu, João Luiz Quagliotti Durigan, Patrícia Azevedo Garcia.

**Software:** Cristiane de Almeida Nagata, Tânia Cristina Dias da Silva Hamu, João Luiz Quagliotti Durigan, Patrícia Azevedo Garcia.

**Supervision:** Tânia Cristina Dias da Silva Hamu, João Luiz Quagliotti Durigan, Patrícia Azevedo Garcia.

**Validation:** Tânia Cristina Dias da Silva Hamu, Paulo Henrique Silva Pelicioni, João Luiz Quagliotti Durigan, Patrícia Azevedo Garcia.

**Visualization:** Cristiane de Almeida Nagata, Tânia Cristina Dias da Silva Hamu, Paulo Henrique Silva Pelicioni, João Luiz Quagliotti Durigan, Patrícia Azevedo Garcia.

**Writing – original draft:** Cristiane de Almeida Nagata, Tânia Cristina Dias da Silva Hamu, João Luiz Quagliotti Durigan, Patrícia Azevedo Garcia.

**Writing – review & editing:** Cristiane de Almeida Nagata, Tânia Cristina Dias da Silva Hamu, Paulo Henrique Silva Pelicioni, João Luiz Quagliotti Durigan, Patrícia Azevedo Garcia.

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
