## [Decision Letter · Decision Letter 0]

26 Dec 2023

PONE-D-23-25022Influence of lower limb isokinetic muscle strength and power on the occurrence of falls in community-dwelling older adults: a longitudinal studyPLOS ONE

Dear Dr. Cristiane de Almeida Nagata,

Thank you for submitting your manuscript to PLOS ONE. After careful consideration, we feel that it has merit but does not fully meet PLOS ONE’s publication criteria as it currently stands. Therefore, we invite you to submit a revised version of the manuscript that addresses the points raised during the review process.

We look forward to receiving your revised manuscript.

Kind regards,

Mehrnaz Kajbafvala, Ph.D

Academic Editor

PLOS ONE

Journal Requirements:

"This study was financed in part by the Postgraduate Department from the University of Brasília (SEI N° 23106.102043/2017-01), the Deanery of research from the State University of Goiás (CCB 01/2018), “Coordenação de Aperfeiçoamento de Pessoal de Nível Superior—Brasil” (CAPES) (Finance Code 001), “Fundação de Apoio a Pesquisa do Distrito Federal” (FAPDF) (grant number 00193-00000866/2023-6, 00193.00000773/2021-72, 00193.00000859/2021-3; 00193.00001222/2021-26), and the National Council for Scientific and Technological Development (CNPq; process numbers 309435/2020-0 and 310269/2021). The funders had no role in study design, data collection and analysis, decision to publish, or preparation of the manuscript."

Reviewers' comments:

Reviewer's Responses to Questions

**Comments to the Author**

1. Is the manuscript technically sound, and do the data support the conclusions?

Reviewer #1: Partly

Reviewer #2: Yes

2. Has the statistical analysis been performed appropriately and rigorously? 

Reviewer #1: No

Reviewer #2: Yes

3. Have the authors made all data underlying the findings in their manuscript fully available?

Reviewer #1: Yes

Reviewer #2: Yes

4. Is the manuscript presented in an intelligible fashion and written in standard English?

Reviewer #1: Yes

Reviewer #2: Yes

5. Review Comments to the Author

Reviewer #1: This study was aimed to evaluate Influence of lower limb isokinetic muscle strength and power on the occurrence of falls in community-dwelling older adults. Authors concluded that hip abductor strength and hip flexor power can be considered protective factors against falls in independent older adults in the community. These findings may help in the development of effective fall-prevention strategies for older adults. Overall, the study is interesting, however there are some clarifications needed.

Comment#1

Abstract, Line 31-33, considering the design of this study and the analyses performed, it seems that the statement of dose-response relationship may not appropriate. Please edit it throughout the manuscript.

Comment#2

Keywords, the keywords should be reflexive of the study performed. Please edit Keywords according to the MESH.

Comment#3

Introduction, the authors should provide more explanations for investigating the influence of strength and power of the all eight muscle groups in the lower limbs on falls in older adults.

Comment#4

Methods, Line 102-106: Authors stated one of the inclusion criteria as "no marked cognitive impairment examined by the Mini-Mental State Examination (MMSE) (cut-off point of 20 points for illiterate individuals; 25 points for people with 1 to 4 years of schooling; 26.5 for 5 to 8 years of schooling; 28 for those with 9 to 11 years of schooling; and 29 for more than 11 years of schooling). Please provide justification for considering these cut-off points.

Comment#5

Discussion, please discuss the significant results obtained in Table 2.

Comment#6

Discussion, the authors should provide the reason/reasons for the lack of significant relationships between strength and power of muscle groups in knee and ankle joints and falls in older adults.

Reviewer #2: The paper was on “Influence of lower limb isokinetic muscle strength and power on the occurrence of falls ”. The article is generally well written and I came to the opinion that this article needs minor revision. I am including my comments and I will like the authors to address the following comments:

Abstract

• The manuscript does need a structured abstract with distinct headings.

• Some more information regarding participants' characteristics.

• More details regarding study results needed.

Introduction

• Page 3, lines 63-65: " …strength decreases by 1.8 to 2.0% per year and power by 3.2 to 3.7% per year. According to Frontera et al., there is a 20 to 30% reduction in the isokinetic strength of knee flexors and extensors at low and high speeds in twelve years". It would have been good the ages that the reductions occur in mentioned studies was highlighted more clearly.

• Page 4, lines 88-93: Objectives and hypotheses of the study must be stated clearly. Please revise.

Methods

• Page 6, line 123: Some details of functional performance tests (sit to stand and walking speed) required.

• Page 7, line 143: Pictures are useful to illustrate muscle strength and power assessment by Biodex system. Can this be included?

Discussion

• Based on the results of study “the fallers showed worse functional performance than non-fallers in the sit-to-stand and walking speed tests” I would suggest that this should be discussed as one of the findings of study in section for discussion. Can this be included?

6. PLOS authors have the option to publish the peer review history of their article (what does this mean?). If published, this will include your full peer review and any attached files.

Reviewer #1: No

Reviewer #2: **Yes: **Amin Behdarvandan

---

## [Author Response · Author response to Decision Letter 0]

16 Feb 2024

Mehrnaz Kajbafvala, Ph.D

Academic Editor

PLOS ONE

February 16, 2024

Dear Editor, 

Revision of the Manuscript Number: PONE-D-23-25022

Title: " Influence of lower limb isokinetic muscle strength and power on the occurrence of falls in community-dwelling older adults: a longitudinal study".

Thank you for the opportunity to resubmit our manuscript. The comments from the reviewers were highly insightful and enabled us to improve the quality of our manuscript. We have changed it according to their advice. The responses to the reviewers are attached, as well as the manuscript's marked-up version (highlighted in yellow).

Thank you again for your time and effort in considering this manuscript for publication. 

Yours sincerely,

Journal Requirements:

Answer: You are very welcome for the review opportunity. We have implemented the requested changes as per the provided instructions.

Answer: We appreciate your valuable suggestion; however, at the moment, we are not interested in pursuing the proposal. All data will be made available upon request to the corresponding author. Thank you for your understanding.

"This study was financed in part by the Postgraduate Department from the University of Brasília (SEI N° 23106.102043/2017-01), the Deanery of research from the State University of Goiás (CCB 01/2018), “Coordenação de Aperfeiçoamento de Pessoal de Nível Superior—Brasil” (CAPES) (Finance Code 001), “Fundação de Apoio a Pesquisa do Distrito Federal” (FAPDF) (grant number 00193-00000866/2023-6, 00193.00000773/2021-72, 00193.00000859/2021-3; 00193.00001222/2021-26), and the National Council for Scientific and Technological Development (CNPq; process numbers 309435/2020-0 and 310269/2021). The funders had no role in study design, data collection and analysis, decision to publish, or preparation of the manuscript."

Answer: Thank you for the observations. We have incorporated the requested changes in the cover letter and in the Funding section of the manuscript, page 19, lines 280 to 290.

“This study was partially funded by the Postgraduate Department from the University of Brasília, https://dpg.unb.br/ (SEI N° 23106.102043/2017-01 author CAN), the Deanery of research from the State University of Goiás, https://ueg.br/prp/ (CCB 01/2018 author TCDSH), “Fundação de Apoio a Pesquisa do Distrito Federal” (FAPDF), https://www.fap.df.gov.br/ (grant number 00193-00000866/2023-6 author PAG, 00193.00000773/2021-72 author JLQD, 00193.00000859/2021-3 author JLQD; 00193.00001222/2021-26 author JLQD; 00193-00001261/2021-23 author JLQD), and the National Council for Scientific and Technological Development (CNPq), https://www.gov.br/cnpq/pt-br (process numbers 309435/2020-0 and 310269/2021 author JLQD). The funders had no role in the study design, data collection and analysis, decision to publish, or preparation of the manuscript. There was no additional external funding received for this study.”

Answer: We apologize for the oversight. The material sent was incorrectly categorized as supporting information. Now, the revised version we submitted it correctly. 

Reviewers' comments:

REVIEWER #1

Reviewer #1: This study was aimed to evaluate Influence of lower limb isokinetic muscle strength and power on the occurrence of falls in community-dwelling older adults. Authors concluded that hip abductor strength and hip flexor power can be considered protective factors against falls in independent older adults in the community. These findings may help in the development of effective fall-prevention strategies for older adults. Overall, the study is interesting, however there are some clarifications needed.

Answer: Thank you for the feedback and the time spent carefully reading our manuscript.

Comment#1

Abstract, Line 31-33, considering the design of this study and the analyses performed, it seems that the statement of dose-response relationship may not appropriate. Please edit it throughout the manuscript.

Answer: Thank you for the feedback. We have incorporated the requested changes throughout the manuscript, specifically in the abstract section, page 2, line 31; introduction section, page 4, line 79; discussion section, page 17, line 238.

Comment#2

Keywords, the keywords should be reflexive of the study performed. Please edit Keywords according to the MESH.

Answer: Thank you for the feedback, however our keywords reflect the main outcomes of the study and are already in accordance with MeSH. Please find below our keywords according to MESH:

Accidental Falls: https://www.ncbi.nlm.nih.gov/mesh/68000058

Muscle Strength: https://www.ncbi.nlm.nih.gov/mesh/68053580

Aged: https://www.ncbi.nlm.nih.gov/mesh/68000368

Comment#3

Introduction, the authors should provide more explanations for investigating the influence of strength and power of the all eight muscle groups in the lower limbs on falls in older adults.

Answer: Thank you sincerely for your consideration and feedback. We appreciate your observations and have incorporated the requested changes in the introduction section, specifically on page 4, from lines 78 to 102. We hope the additional explanations regarding the influence of strength and power of all eight muscle groups in the lower limbs on falls in older adults are clearer. 

“While the association between muscle strength, muscle power, and falls has been demonstrated in previous studies [1–14], the specific influence of each lower limb muscle group's strength and power on the occurrence of falls in older individuals remains unclear. Some studies suggest that ankle muscle strength and power are more closely associated with falls in older adults [1,2,7–11,15]. However, other research indicates that knee flexors [12,13], knee extensors [12–14], hip flexors [6], hip extensors [6,13,14], hip abductors [6,12], and hip adductors [6] also play crucial roles in this outcome.

Notably, none of the cited studies have comprehensively examined the influence of all eight muscle groups in the lower limbs on falls in older adults. Additionally, most of the available studies did not evaluate muscle strength and power using the isokinetic dynamometer [2,7,9,12] (considered the gold standard for assessing muscle strength [16]), and they often focused on isolated muscle groups or a subset of the lower limb muscles assessment [1,2,6–8,10,13], which may not provide a complete understanding of overall muscle function.

Therefore, understanding the simultaneous association between all eight muscle groups of the lower limbs and falls in older adults is crucial for developing an effective fall prevention program. Our study addresses this gap by investigating by isokinetic dynamometer the influence of ankle plantarflexors, ankle dorsiflexors, knee flexors, knee extensors, hip flexors, hip extensors, hip abductors and hip adductors muscle strength and muscle power on falls in older adults, aiming to determine which muscle groups are more predictive of fall risk in this population. This comprehensive approach may offer insights for developing targeted and effective strategies in fall prevention and rehabilitation. Based on the existing literature [1–15], our hypothesis is that our findings will align with previous findings[1,2,7–11,15], indicating that ankle muscles will be more predictive of the occurrence of falls in older adults.”

Comment#4

Methods, Line 102-106: Authors stated one of the inclusion criteria as "no marked cognitive impairment examined by the Mini-Mental State Examination (MMSE) (cut-off point of 20 points for illiterate individuals; 25 points for people with 1 to 4 years of schooling; 26.5 for 5 to 8 years of schooling; 28 for those with 9 to 11 years of schooling; and 29 for more than 11 years of schooling). Please provide justification for considering these cut-off points.

Answer: Thank you for your question. We have adopted the cutoff points for cognitive impairment assessment established by the Mini-Mental State Examination (MMSE), based on the recent research conducted by Brucki et al., 2003 (https://doi.org/10.1590/S0004-282X2003000500014). This research was conducted in the Brazilian context, and the criteria employed are specifically tailored to the reality of our sample, comprising Brazilian older adults. We selected these criteria to guarantee a more precise assessment of cognitive impairment, considering variations in educational levels within our sample. This approach is designed to better align with cognitive patterns in the target population, ensuring the validity and representativeness of the study results.

Comment#5

Discussion, please discuss the significant results obtained in Table 2.

Answer: Thank you for the consideration. We discussed it on page 17, lines 254-261:

“Comparisons between older adults who experienced a fall and those who did not revealed that non-fallers had significantly better isokinetic muscle strength in the hip abductors and adductors, as well as higher muscle power in hip abductors, hip flexors, and knee flexors compared to fallers. This finding is somewhat surprising, considering that several previous studies suggested that ankle muscle strength and power are more closely associated with falls in older adults [1,2,7–11,15]. On the other hand, Morcelli et al. [6] had already demonstrated that older fallers had significantly lower strength in hip extension, abduction, and adduction, as well as lower power in hip flexion, extension, and abduction compared to non-fallers, which is consistent with our findings.”

Comment#6

Discussion, the authors should provide the reason/reasons for the lack of significant relationships between strength and power of muscle groups in knee and ankle joints and falls in older adults.

Answer: Thank you for the consideration. We discussed it on page 18, lines 286-296: 

“Contrasting with findings from prior studies [1,2,7–15], our observations highlight a notable absence of significant associations between the strength and power of muscle groups in the knee and ankle joints and occurrences of falls in older adults. This discrepancy can be attributed to the fact that previous research did not include a comprehensive analysis of all eight muscle groups in the lower limbs and did not utilize the isokinetic dynamometer, acknowledged as the gold standard for assessing muscle strength [16]. These methodological limitations likely played a role in the disparities among study outcomes, posing a challenge in establishing coherent relationships between distinct muscle groups and the incidence of falls. Additionally, most research has focused exclusively on assessing the musculature of the ankle and/or knee, disregarding the analysis of hip muscles. This may explain why the importance of these muscles in functional performance and fall prevention in older adults has not been fully elucidated.”

REVIEWER #2

Reviewer #2: The paper was on “Influence of lower limb isokinetic muscle strength and power on the occurrence of falls”. The article is generally well written and I came to the opinion that this article needs minor revision. I am including my comments and I will like the authors to address the following comments:

Answer: We sincerely appreciate the time and effort invested by the reviewer in critically assessing our manuscript. The insightful feedback and constructive comments have significantly contributed to the improvement of our work. 

Abstract

• The manuscript does need a structured abstract with distinct headings.

Answer: Thank you for your feedback. We have introduced the distinct headings in the abstract.

• Some more information regarding participants' characteristics.

Answer: Thank you for the observation. We have introduced more information regarding participants' characteristics. Page 2, Lines 39-41:

“Results: Participants, with a median age of 69 years (range 64-74), included 67% women, and 63.8% reported a sedentary lifestyle. Among them, 45,7% of older adults were classified as fallers.”

• More details regarding study results needed.

Answer: Thank you for the observation. We have introduced more details regarding study results. Page 2, Lines 39-44:

“Results: Participants, with a median age of 69 years (range 64-74), included 67% women, and 63.8% reported a sedentary lifestyle. Among them, 45,7% of older adults were classified as fallers. Comparative analyses revealed that non-fallers displayed significantly superior isokinetic muscle strength in the hip abductors and adductors, along with higher muscle power in the hip abductors, hip flexors, and knee flexors compared to fallers.”

Introduction

• Page 3, lines 63-65: " …strength decreases by 1.8 to 2.0% per year and power by 3.2 to 3.7% per year. According to Frontera et al., there is a 20 to 30% reduction in the isokinetic strength of knee flexors and extensors at low and high speeds in twelve years". It would have been good the ages that the reductions occur in mentioned studies was highlighted more clearly.

Answer: Thank you for the feedback. We have introduced this information on Page 3, Lines 65-67:

“Muscle strength decreases by 1.8 to 2.0% per year and power by 3.2 to 3.7% per year in older adults (age range 65-89 years) [17]. According to Frontera et al. [18], in sedentary men with initial mean age 65.4 years-old (age range 60-72 years), there was a 20 to 30% reduction in the isokinetic strength of knee flexors and extensors at low and high speeds in twelve years.”

• Page 4, lines 88-93: Objectives and hypotheses of the study must be stated clearly. Please revise.

Answer: Thank you for your feedback. After carefully examining the literature, we added our hypothesisas follows:

Page 4, lines 100 to 102: “Based on the existing literature[1–15], our hypothesis is that our findings will align with previous findings[1,2,7–11,15], indicating that ankle muscles will be more predictive of the occurrence of falls in older adults.”

Methods

• Page 6, line 123: Some details of functional performance tests (sit to stand and walking speed) required.

Answer: Thank you for the feedback. We have introduced this information on Page 6, Lines 137-144:

 “To assess walking speed, a 4-meter distance was marked, and participants were instructed to walk at their usual pace. The time to complete the test was recorded in two attempts, with the shorter time to complete the walking assessment considered for scoring. The usual walking speed (m/s) was then calculated by dividing the distance covered in the test (4 meters) by the shortest time recorded for that course in seconds [19–21]. In the five-time sit-to-stand test, participants were asked to rise from a chair, starting from a seated position, with arms crossed over the chest, repeating the movement five times as quickly as possible. The elapsed time in seconds was recorded as a performance measure [19–21].”

• Page 7, line 143: Pictures are useful to illustrate muscle strength and power assessment by Biodex system. Can this be included?

Answer: Thank you very much for the observations. We have inserted a figure to illustrate muscle strength and power assessment by the Biodex system on “Materials and Methods” section, Page 9, lines 188-191.

Discussion

• Based on the results of study “the fallers showed worse functional performance than non-fallers in the sit-to-stand and walking speed tests” I would suggest that this should be discussed as one of the findings of study in section for discussion. Can this be included?

Answer: Thank you for the feedback. We have discussed these results on Page 17, Lines 240-253:

“Our findings indicated that fallers exhibited worse functional performance than non-fallers in the sit-to-stand and walking speed tests. These findings align with various prior studies [22–27] that emphasized the significant association between slower walking speed and occurrences of falls in older adults. These results can be explained by the impact of aging on the gait of older individuals [28], characterized by changes in three-dimensional kinetics and kinematics, including reduced movement of the pelvis in the frontal and transverse planes, prolonged hip adduction, and increased knee extension peak during the support phase (hyperextension). These alterations, coupled with reduced ankle plantar flexion during the terminal support phase and decreased knee range of motion compared to younger individuals, contribute to the decrease in walking speed [28]. Additionally, previous studies have also highlighted the association between poorer performance in the sit-to-stand test and a higher risk of falls in older adults [29–31]. This relationship can be attributed to the influence of aging on this function [17,32], as evidenced by the greater difficulty that older individuals face when rising from a chair compared to younger counterparts [32,33].”

---

## [Decision Letter · Decision Letter 1]

6 Mar 2024

Influence of lower limb isokinetic muscle strength and power on the occurrence of falls in community-dwelling older adults: a longitudinal study

PONE-D-23-25022R1

Dear Dr. Cristiane de Almeida Nagata

We’re pleased to inform you that your manuscript has been judged scientifically suitable for publication and will be formally accepted for publication once it meets all outstanding technical requirements.

Kind regards,

Mehrnaz Kajbafvala, Ph.D

Academic Editor

PLOS ONE

Additional Editor Comments (optional):

Reviewers' comments:

Reviewer's Responses to Questions

**Comments to the Author**

1. If the authors have adequately addressed your comments raised in a previous round of review and you feel that this manuscript is now acceptable for publication, you may indicate that here to bypass the “Comments to the Author” section, enter your conflict of interest statement in the “Confidential to Editor” section, and submit your "Accept" recommendation.

Reviewer #1: All comments have been addressed

Reviewer #2: All comments have been addressed

2. Is the manuscript technically sound, and do the data support the conclusions?

Reviewer #1: Yes

Reviewer #2: Partly

3. Has the statistical analysis been performed appropriately and rigorously? 

Reviewer #1: Yes

Reviewer #2: Yes

4. Have the authors made all data underlying the findings in their manuscript fully available?

Reviewer #1: Yes

Reviewer #2: Yes

5. Is the manuscript presented in an intelligible fashion and written in standard English?

Reviewer #1: Yes

Reviewer #2: Yes

6. Review Comments to the Author

Reviewer #1: (No Response)

Reviewer #2: (No Response)

7. PLOS authors have the option to publish the peer review history of their article (what does this mean?). If published, this will include your full peer review and any attached files.

Reviewer #1: No

Reviewer #2: **Yes: **Amin Behdarvandan

---

## [Editor Report · Acceptance letter]

26 Mar 2024

PONE-D-23-25022R1 

PLOS ONE

Dear Dr. de Almeida Nagata, 

I'm pleased to inform you that your manuscript has been deemed suitable for publication in PLOS ONE. Congratulations! Your manuscript is now being handed over to our production team.

Kind regards, 

on behalf of

Dr. Mehrnaz Kajbafvala 

Academic Editor

PLOS ONE